# The Effect of Religious Beliefs and Attitudes in Intrinsic and Extrinsic Optimism and Pessimism in Players of Games of Chance

Lisete S. Mónico * and Valentim R. Alferes

Faculty of Psychology and Education Sciences, University of Coimbra, 3000-115 Coimbra, Portugal;
valferes@fpce.uc.pt
* Correspondence: lisete.monico@fpce.uc.pt

**Abstract:** Games of chance usually make people feel a whirlwind of emotions, especially in gambling. While those games depend more on luck than on individuals' skills, optimism should be a distinctive feature. Considering the classic literature of the effects of religiosity on risk behaviors, the issue of the influence of religiosity on optimism in players of games of chance has been less studied, especially when we considered optimism as a multidimensional concept comprising intrinsic and extrinsic optimism and pessimism. Aims: To analyze the effect of religious beliefs and attitudes in optimism and pessimism dimensions in players of games of chance and gambling. Method: The sample consists of 271 recurring players of games of chance and gambling, who answered a questionnaire composed of measures of religious beliefs and attitudes, optimism, pessimism, and estimates of future occurrences, evidencing good psychometric properties. Results: Players are moderately religious and more optimistic than pessimistic, estimating a chance of 36% of highly unlikely desirable events. The structural model showed an overall influence of religious beliefs and attitudes higher on optimism ($R^2 = 44\%$) than on pessimism ($R^2 = 5\%$). However, the distinction between intrinsic and extrinsic optimism has shown that the players anchor their optimism in different kinds of beliefs. Extrinsic desirable events, like winning the lottery, were more predicted by religious beliefs and attitudes in comparison with intrinsic desirable events. Inversely, religious beliefs and attitudes tend to predict more intrinsic pessimism in comparison with intrinsic optimism. Conclusions: Optimism is not a one-dimensional construct, should be analyzed considering the dichotomies of optimism/pessimism and intrinsic/extrinsic. In recurring players of games of chance and gambling, religious beliefs and attitudes predicted more optimism than pessimism, being more associated with extrinsic than intrinsic desirable events. More intrinsically pessimistic players seem to recur to religiosity to anchor their positive expectations.

**Keywords:** religious beliefs; religious attitudes; optimism; pessimism; players of games of chance

## 1. Introduction

When we think about players of games of chance and gambling, the idea of optimistic people comes to mind. Games of chance usually make people feel a whirlwind of emotions, especially in gambling. Whereas those games depend more on luck than on individuals' skills, optimism should be a distinctive feature.

Whenever a person plays, they have at least a glimmer of hope that they can win. Following this reasoning, players should be optimists or, at least, more optimistic than pessimists. Most of the literature shows that people, in general, perceive their future as being happier than the future of other people, believing that they are more likely to experience desirable situations and less likely to experience undesirable ones (Mens et al. 2021).

Considering that all areas of life are mediated by aims, the behavior of individuals is determined by the self-regulatory mechanisms adopted with a view to achieving them.

Predictions based on the theory of social comparison (Festinger 1954) would not give optimism its general character (Carver and Scheier 2001; Mónico 2021). Furthermore, in the course of the life cycle, the emergence of adversities is practically inevitable, and the difficulty in overcoming them may lead individuals to pessimism (Swann et al. 1987). In the case of players of games of chance and gambling, the times they lost a game or a bet are very frequent, and the reasoning concerning the chances of winning should point to low probabilities. For Carver and Scheier (1982, 2012), optimism enters into self-regulation when people, despite anticipating obstacles to achieving certain goals, maintain the belief that they will be successful (Armor and Taylor 1998; Scheier and Carver 1992). The aim of this paper is to analyze if and how optimism can be anchored in religious beliefs and attitudes in players of games of chance and gambling.

Classic is the idea that religion instigates, among other functions, normative behavior. However, the investigation that religion can promote optimism has been less studied, especially when we considered optimism as a multidimensional concept comprising intrinsic and extrinsic optimism and pessimism. It is not uncommon to observe situations in which players of games of chance resort in some way to their religiosity, believing that it will help them to win. In Portugal, some research has identified a positive association between religiosity and optimism (Mónico 2012a, 2013a, 2013b; Mónico and Alferes 2019; Mónico et al. 2016). Classic authors also pointed some connections between religiosity and some dimensions of psychological capital (e.g., W. James, Freud, Weber, Durkheim, Allport; Mattis et al. 2004). It is interesting to explore this relationship, especially if and how players anchor their optimism in their religiosity, and to what extent optimism and pessimism dimensions are associated with a perception of greater probability of occurrence of desirable events and the prevention of the undesirable ones.

## 2. Background

### 2.1. Religiosity and Religious Beliefs and Attitudes

It is understood that religiosity, or religious culture (Sitzmann and Campbell 2021), is the individual level of commitment to beliefs, doctrines, and practices of some religion (Barker and Warburg 1998; Mookherjee 1994). Counterpart expression of religious experience (Geerts 1990) concerns the extent to which an individual believes, follows, and practices a religious doctrine, considering its two regulating poles: beliefs and rites. In the classic work of (James [1902] 1985), religiosity is defined as "the feelings, acts, and experiences of individual men in their solitude, so far as they apprehend themselves to stand in relation to whatever they may consider the divine" (p. 34). This can be introduced either in a traditional way, in a formal and non-reflective way that follows the customs, or in an individual way, looking for answers to questions, needs, ideas, and ideals (Grom 1994).

People's religiosity is highly influenced by culture (Sitzmann and Campbell 2021), religious practices, and motivations. Within and between religious groups, the nature and intensity of beliefs are extremely variable (Ávila 2003; Mónico 2011; Pargament 1997). Multiple surveys have been carried out with the aim of ascertaining the extent to which people hold religious beliefs (Hinde 2010). However, the meaning of the expression "I believe" becomes controversial and difficult to ascertain (Gellner 1992). The boundaries that distinguish faith and belief are not clear. Moreover, spirituality is independent of any religion or belief system, considered as a complex multi-dimensional and multi-cultural concept (Mónico and Margaça 2021).

Although it has different meanings (Fowler 1995; Hood 1995; Pargament 1997; Wulff 1997), the construct of faith is indistinguishable from that of other attitudes . Focusing on the reasoning and dynamic process of elaboration, faith is seen as an adherence of the mind founded on arguments that do not constitute a rigorous demonstration (...), a mental attitude that includes both a commitment and a free adhesion" (p. 92).

Argyle (2000) equates faith with an attitude that, as a favorable or unfavorable disposition, expressed in words and/or behavior (Eagly and Chaiken 1998), can be divided

into the classical cognitive, emotional, and behavioral components. Hinde (2010) highlights the supernatural focus of religious beliefs, as they involve unusual beings, entities, and experiences, encompassing counter-intuitive pretensions and complex concepts, not fully intelligible and often controversial or inconsistent. In fact, religious beliefs are not constrained to the possibility of empirical materialization (Haught 1995), are established by authority, by consensus, or by both (Fowler 1995; Lawson and McCauley 1990), and they are supported by social consent and traditions (Brown 1988). Thus, we understand the tendency in modern western societies to consider religious beliefs as mere opinions or attitudes, as opposed to empirical beliefs seen as knowledge. However, we are in line with McGuire's (2002) conception, which considers that both beliefs—religious and empirical—constitute "knowledge" for the individual who believes in them, being real in their consequences and outlining the experiences and actions of the individual.

The double meaning of religious beliefs and attitudes is pointed out by the classic work of Dittes (1969), when he states that the individual believes in a supernatural or superhuman objective reality, however, based on the subjectivity of the psychological conditions of human beings. According to the author, for a religious individual, believing is not a way of facing the world and the future, but a relationship with a being/entity through symbolic actions, supported by reports and representations of the divine and inspiring rules of conduct.

### 2.2. Optimism and Pessimism

*Optimism*. The scope of optimism is represented in the literature by two interrelated concepts: the positive expectations for the future (Domino and Conway 2001; Erthal et al. 2021) and the tendency to believe that the world is the "best of all possible worlds" (Gillham et al. 2001, p. 53). "Optimists are people who expect good experiences in the future. Pessimists are people who expect bad experiences" (Carver and Scheier 2001, p. 31). Thus, "optimism is seen as a cognitive feature (a goal, an expectation, a belief or a causal attribution) about the desired and perceived as successful future" (Barros 2004, p. 101). The tendency towards the positive, the expectation of obtaining good results and the explanation attributed to the negative events characterize, in general, optimism, detected in areas of life as distinct as health, professional or academic achievement, interpersonal relationships, and security (Buunk 2001; McKenna 1993; Mónico 2013a, 2021; Mónico et al. 2016; Weinstein 1987). The conceptual definitions are directed towards positive expectations, usually generalized and stable (Mónico 2011, 2021), linked to two key brain areas: the anterior cingulate cortex (ACC—imagination of the future and self-referential information procession) and the inferior frontal gyrus (IFG—response inhibition and handling with important cues). ACC action was positively associated with trait optimism and with the estimations of positive events, and IFG with behavioral measures of optimistic propensity (Erthal et al. 2021).

*Pessimism*. The nature and intensity of beliefs related to the failure to reach the intended goals, essentially in situations of adversity, constitutes an identifier of people's level of pessimism (Mónico 2013c). Absolute or dispositional pessimism refers to generalized expectations of the occurrence of negative events, for the individual, or the tendency to expect unfavorable life outcomes (Kruger and Burrus 2004). A state of pessimism leads to the undertaking of reduced efforts in the achievement of the goals, especially when pessimism is a dispositional trait (Scheier and Carver 1985). The stable tendency to maintain negative expectations about own results reveals a pessimistic trait (Carver and Scheier 2001), consistently influencing expectations throughout situations.

As we find unrealistic optimism in people (Weinstein 1980, 1987), Kruger and Burrus (2004) call attention to the existence of unrealistic pessimism. This type of pessimism occurs for very rare desirable events (e.g., living after 100 years), characterized by the lower expectations of these events for the person, compared to other individuals. In addition to the evidence of comparative optimism, Chambers et al. (2003) propose the existence of a comparative pessimism, detected in desirable and unusual situations, although also

in undesirable and common events. The concept of defensive pessimism was proposed by Cantor et al. in the mid-eighties and represents a cognitive strategy that individuals use to prepare for stress-inducing situations (Norem and Cantor 1986), differing from the attributional style defensive of Seligman (Gillham et al. 2001; Seligman 2006) and the pessimism-trait advocated by Carver and Scheier (2001). The question is whether it is always adaptive to expect the best (Carver and Scheier 2001).

*Optimism, pessimism, and gambling.* Some of the literature has been devoted to studying the relationship between optimism and gambling. Gibson and Sanbonmatsu (2004) found that optimistic players were more likely to have positive gambling expectations and report maintaining these expectations following losses, in comparison with pessimistic players. They also indicated money as the main motivation for gambling. After poor gaming performance, the pessimistic players tend to decrease more their betting and expectations, when compared to optimists. These last players, after losing, recalled more wins than do pessimistic players.

*Intrinsic* vs. *extrinsic optimism and pessimism.* In this paper, we consider the distinction between intrinsic and extrinsic optimism and pessimism (Mónico 2011, 2012b, 2013c). Intrinsic optimism refers to the expectation that good future experiences depend on their own personal skills and extrinsic optimism to the conviction that the good results will prevail due to situational factors, not having the elderly control over these factors (luck, chance).

A basic premise of optimism anchored in internality beliefs is the expectation that desirable occurrences will happen via assignment of causality to factors internal to the individual, personal, and dependent of himself. Inversely, individuals with optimism based on externality beliefs believe that their positive events will be determined by situational factors, external and not controllable by themselves, caused by others, or determined by luck or by chance.

Applying the concept of internality and externality to pessimism, we found the same reasoning. As the locus of control (Rotter 1990), we consider that the continuum which goes from extreme optimism to extreme pessimism is permeated by internality or externality beliefs, and the anticipation of positive (optimism) or negative (pessimism) outcomes can be attributed to internal or external individual factors. Thus, by internality optimism, we consider the expectation that good future experiences depend on their own personal skills. Externality optimism refers to the conviction that good results will prevail due to situational factors, with the elderly control not having over these factors, like luck or chance (Mónico 2011, 2012b, 2013c). In this research, in addition to measures of optimism and pessimism based on conventional authors, we operationalized optimism and pessimism based on the estimation of the occurrence of certain events in the respondent's life, both positive and negative, based either on intrinsic or extrinsic factors.

## 3. Method

### 3.1. Sample

The sample consisted of 271 Portuguese recurring players of games of chance and gambling, 186 (68.6%) being male and 85 (31.4%) male, with an average age of 41.50 years-old (SD = 14.97; age range: 16–87 years), $M_{age}$ = 42.72 (SD = 15.55) for males and $M_{age}$ = 38.85 (SD = 13.32) for females. Regarding education, 44 (16.2%) participants completed 4 years of education, 72 (26.6%) 9 years of education, 93 (34.3%) 12 years of education, and 62 (22.9%) completed higher education. The majority of the participants were married (n = 166, 61.3%), 83 (30.6%) is single, 17 (6.3%) divorced, and 4 (1.5%) are widowed (1 missing-value).

In total, 58 players (21.4%) lived in the countryside, 73 in a suburban area (26.9%), and 139 in urban areas (51.3%) (1 m:L:ssing-value, 0.4%); 259 belonged to Portugal Continental (95.6%), 62 (22.9%) to the north of the country, 169 (62.4%) to the central region, 21 (7.7%) to Lisboa and Vale do Tejo, 5 (1.8%) to the Alentejo, 2 (0.7%) to the Algarve, and 12 (84.4%) to Portuguese islands Madeira and Azores. With regard to the professional situation of

the respondents, the majority were employed (n = 225, 83.0%), with diverse occupations, followed by students (n = 32, 11.8%) and retired (n = 14, 5.2%).

### 3.2. Data Analysis

All the analysis was performed by using the statistical program SPSS and AMOS (IBM Corp. 2020). Skewness and kurtosis values indicate a normal distribution, |Sk| < 1.30 and |Ku| < 1.73 ($-0.562$ < Sk < 0.807 and $-0.592$ < ku < 0.683 for the composite scores).

Exploratory Factor Analysis (EFA) was performed with SPSS by Principal Component Analysis (PCA), VARIMAX rotation (Kaiser's normalization), given that we expected independent factors. The PCA assumptions were tested through the sample size (ratio of 5 subjects per item and at least 100 participants; Gorsuch 2015), the normality and linearity of the variables, factorability of *R*, and sample adequacy (Tabachnick and Fidell 2019). Reliability was calculated by Cronbach's alpha (Nunnally and Bernstein 2010). The score of 0.80 was taken as a good reliability indicator (Urbina 2014), and 0.60 as acceptable (DeVellis 2012).

For the analysis of variance (ANOVA) and multivariate analysis of variance (MANOVA), the assumptions of independence of observations and homogeneity of error variance and covariance matrices of the dependent variables were checked. Post hoc Tukey HSD tests were performed for pairwise multiple comparisons.

Structural equation modeling was carried out with IBM AMOS and the maximum likelihood estimation method. The goodness of fit was analyzed using CMIN/DF (normed chi-square), NFI (normed fit index), CFI (comparative fit index), and RMSEA (Root Mean Square Error of Approximation) (Kline 2016; Schumacker and Lomax 2016).

Internal consistency was assessed by Cronbach's alpha coefficient (Nunnally and Bernstein 2010), both for the global scale and their dimensions. Despite reliability coefficients higher than 0.70 being considered acceptable for convergence and reliability, we have based on Nunnally and Bernstein (2010) and DeVellis (2012) for reliability in each dimension. Mean scores were calculated based on the average of items in each factor.

A probability of 0.05 for the Type I error was considered for all the inferential statistics.

### 3.3. Materials

A survey was carried out using a self-administered questionnaire composed of the Religious Beliefs and Attitudes Scale, the Optimism and Pessimism Scale, and a sociodemographic questionnaire.

#### 3.3.1. Religious Beliefs and Attitudes Scale

This scale was built and validated by Mónico (2011) with a larger sample of Portuguese citizens. It is composed of 13 multiple-choice items (from 1 = totally disagree to 5 = totally agree). The PCA performed with this sample (see Table 1) pointed to a one-factor solution responsible for 79.74% of the total variability and good reliability ($\alpha$ = 0.96, see Table 1).

#### 3.3.2. Optimism and Pessimism Scale

The *Optimism and Pessimism Scale* were adapted from the literature (Barros 1998), Scheier et al. (1994), Schweizer and Koch (2001), Snyder et al. (1991) and Wiseman (2006). The 12 multiple-choice items, answered from 1 (strongly disagree) to 5 (strongly agree), were analyzed through a PCA (see Table 2), emerging two independent factors with acceptable reliability: *Optimism* (7 items, $\alpha$ = 0.71) and *Pessimism* (5 items, $\alpha$ = 0.66).

**Table 1.** *Religious Beliefs and Attitudes Scale (RBAS*: mean scores (M), standard-deviations (SD), factorial loadings (s), commonalities ($h^2$), and Cronbach's internal consistency coefficient ($\alpha$) for the one-dimension solution.

| | | Items | M | SD | s | $h^2$ |
|---|---|---|---|---|---|---|
| [DEUS_36] | [RBA1] | I believe God hears my prayers. | 3.46 | 1.32 | 0.88 | *0.77* |
| [DEUS_13] | [RBA2] | In moments of happiness, I believe it was God who helped me. | 3.44 | 1.31 | 0.88 | *0.78* |
| [DEUS_19] | [RBA3] | I feel that God protects me from the adversities of life. | 3.32 | 1.27 | 0.90 | *0.81* |
| [DEUS_24] | [RBA4] | I need God's help to make important decisions in my life. | 2.93 | 1.36 | 0.87 | *0.76* |
| [DEUS_40] | [RBA5] | Everything I am and everything I hope to be I owe to God. | 2.93 | 1.29 | 0.85 | *0.72* |
| [DEUS_34] | [RBA6] | I usually thank God for the happiness of my life. | 3.49 | 1.32 | 0.85 | *0.73* |
| [DEUS_32] | RBA7 | When I have a problem, I get closer to God. | 3.34 | 1.26 | 0.79 | *0.62* |
| [DEUS_28] | RBA8 | The universe was created by God. | 3.42 | 1.44 | 0.77 | *0.59* |
| [DEUS_31] | RBA9 | I trust what God has destined for me. | 3.38 | 1.30 | 0.86 | *0.74* |
| [DEUS_37] | RBA10 | I believe that God will reward me for my current sufferings. | 3.26 | 1.28 | 0.78 | *0.60* |
| [DEUS_58] | RBA11 | Without faith in God, I would lose the strength to fight. | 3.08 | 1.35 | 0.79 | *0.63* |
| [DEUS_39] | RBA12 | Lately, my belief in God has increased. | 2.95 | 1.28 | 0.77 | *0.59* |
| [DEUS_21] | RBA13 | I trust God more than myself to overcome problems. | 2.41 | 1.25 | 0.70 | *0.49* |
| | TOTAL KMO = 0.97; Bartlett's test: $\chi^2$ (78) = 3082.28 ($p < 0.001$); $\alpha$ = 0.960 | | 3.16 | 1.39 | | |

**Table 2.** PCA of the *Optimism and Pessimism Scale*: Descriptive statistics (*M* and *SD*), factorial loadings (*s*) of the rotatex component matrix (F1, F2), commonalities ($h^2$), eigenvalues, shared variance, and Cronbach's alpha.

| | | | M | SD | F1 (s) | F2 (s) | $h^2$ |
|---|---|---|---|---|---|---|---|
| | | **F1: Optimism** | | | | | |
| 7.44 | IO1 | I vigorously pursue my goals. | 3.86 | 0.84 | **0.67** | −0.04 | 0.45 |
| 7.29 | IO2 | I always find a solution to a problem. | 3.45 | 0.83 | **0.65** | 0.05 | 0.43 |
| 7.53 | IO3 | I have a lot of confidence in myself. | 3.97 | 0.80 | **0.61** | −0.14 | 0.38 |
| 7.26 | IO4 | No task is too difficult for me. | 3.17 | 0.93 | **0.59** | 0.05 | 0.35 |
| 7.62 | IO5 | I am always optimistic about my future. | 3.63 | 0.92 | **0.57** | −0.33 | 0.43 |
| 7.41 | IO6 | I can think of many ways to get out of trouble. | 3.63 | 0.86 | **0.55** | 0.02 | 0.31 |
| 7.23 | IO7 | I overcome even the most difficult problems. | 3.57 | 0.84 | **0.55** | −0.02 | 0.30 |
| | | **F2: Pessimism** | | | | | |
| 7.20 | IP1 | I rarely expect things to go my way. | 2.51 | 1.07 | −0.04 | **0.72** | 0.52 |
| 7.17 | IP2 | When life is going well, I am afraid that there will soon be some adversity. | 2.91 | 1.18 | 0 | **0.65** | 0.43 |
| 7.22 | IP3 | If something can go wrong for me, it sure will happen. | 2.36 | 0.98 | −0.01 | **0.64** | 0.41 |
| 7.35 | IP4 | I rarely hope that good things will happen to me. | 2.46 | 1.08 | −0.01 | **0.61** | 0.37 |
| 7.61 | IP5 | In difficult situations, I am always expecting the worst. | 2.65 | 1.09 | −0.17 | **0.61** | 0.40 |
| | Eigenvalues | | | | 2.76 | 2.00 | |
| | % of explained variance | | | | 23.01 | 16.71 | |
| | Cronbach's $\alpha$ | | | | 0.71 | 0.66 | |

### 3.3.3. Estimation of Future Desirable Events Scale

We considered the estimation of 11 future desirable events, measured from 0 to 100%, adapted from Wiseman (2006). The PCA performed identified two dimensions with good

reliability (see Table 3): *Intrinsic desirable events* (6 items, α = 0.83) and *Extrinsic desirable events* (5 items, α = 0.77).

**Table 3.** PCA of the *Estimation of Desirable Events scale*: Descriptive statistics (*M* and *SD*), factorial loadings (*s*) of the rotatex component matrix (F1, F2), commonalities ($h^2$), eigenvalues, shared variance, and Cronbach's alpha.

| | From 0% to 100%, Please Indicate the Percentage That Best Represents the Possibility of Occurrence of This Event in Your Life . . . | *M* (%) | *SD* | F1 (*s*) | F2 (*s*) | $h^2$ |
|---|---|---|---|---|---|---|
| | **F1: Intrinsic desirable events** (probability 0–100%) | | | | | |
| IDE1 | Having harmony in the family. | 75.26 | 24.78 | **0.84** | 0.08 | 0.72 |
| IDE2 | Being reciprocated in a romantic relationship. | 73.19 | 27.71 | **0.80** | −0.02 | 0.65 |
| IDE3 | Living happily. | 70.74 | 25.46 | **0.79** | 0.29 | 0.70 |
| IDE4 | Get lucky in life. | 58.14 | 26.55 | **0.61** | 0.48 | 0.60 |
| IDE5 | Be strong/have courage. | 64.53 | 26.92 | **0.58** | 0.32 | 0.44 |
| IDE6 | Overcoming my biggest difficulty. | 54.01 | 26.28 | **0.49** | 0.35 | 0.36 |
| | **F2: Extrinsic desirable events** (probability 0–100%) | | | | | |
| EDE1 | Become a millionaire. | 40.65 | 32.63 | 0.11 | **0.81** | 0.67 |
| EDE2 | Win the lottery. | 45.02 | 35.44 | 0.18 | **0.78** | 0.65 |
| EDE3 | Be famous. | 20.90 | 26.75 | 0.14 | **0.65** | 0.45 |
| EDE4 | A miracle happens in my life. | 41.59 | 27.28 | 0.14 | **0.65** | 0.44 |
| EDE5 | Be admired by other people. | 31.96 | 31.44 | 0.28 | **0.55** | 0.38 |
| | Eigenvalues | | | 4.52 | 1.52 | |
| | % of explained variance | | | 28.04 | 26.90 | |
| | Cronbach's α | | | 0.83 | 0.77 | |

### 3.3.4. Estimation of Future Undesirable Events Scale

We also asked participants for the estimation of 14 future undesirable events, measured from 0 to 100%, adapted from Wiseman (2006). The PCA performed identified two dimensions with good reliability (see Table 4): *Intrinsic desirable events* (6 items, α = 0.83) and *Extrinsic desirable events* (5 items, α = 0.77).

**Table 4.** PCA of the *Estimation of Undesirable Events Scale*: Descriptive statistics (*M* and *SD*), factorial loadings (*s*) of the rotatex component matrix (F1, F2), commonalities ($h^2$), eigenvalues, shared variance, and Cronbach's alpha.

| | From 0% to 100%, Please Indicate the Number (in Percentage) That Best Represents the Possibility of Occurrence of This Event in Your Life . . . | *M* (%) | *SD* | F1 (*s*) | F2 (*s*) | $h^2$ |
|---|---|---|---|---|---|---|
| | **F1: Extrinsic undesirable events** (probability 0–100%) | | | | | |
| EUE1 | Having a serious chronic illness. | 45.24 | 28.09 | **0.86** | 0.09 | 0.75 |
| EUE2 | Having a malignant disease (eg,, cancer). | 46.0 | 29.4 | **0.86** | 0.26 | 0.80 |
| EUE3 | Having a cardiovascular disease (eg., heart attack, stroke). | 42.9 | 28.0 | **0.78** | 0.22 | 0.66 |
| EUE4 | Dying soon. | 38.3 | 28.0 | **0.66** | 0.29 | 0.52 |
| EUE5 | Going through difficult times in life. | 46.60 | 26.57 | **0.62** | 0.38 | 0.52 |
| EUE6 | Having a serious accident (eg,, driving, at work). | 43.1 | 28.0 | **0.60** | 0.39 | 0.52 |
| EUE7 | Losing the love of your life (through death, divorce, separation). | 38.1 | 30.9 | **0.54** | 0.36 | 0.42 |
| | **F2: Intrinsic undesirable events** (probability 0–100%) | | | | | |
| IUE1 | Not achieving what I idealize. | 34.70 | 24.63 | 0.19 | **0.73** | 0.57 |
| IUE2 | Not being able to fulfill my duties. | 28.73 | 25.76 | 0.30 | **0.67** | 0.54 |
| IUE3 | Having bad luck in life. | 36.64 | 26.91 | 0.30 | **0.66** | 0.53 |
| IUE4 | Losing hope/becoming a pessimist. | 24.08 | 23.63 | 0.15 | **0.66** | 0.46 |
| IUE5 | Having a worse life than others. | 26.90 | 25.02 | 0.13 | **0.63** | 0.41 |
| IUE6 | Go into depression. | 28.89 | 28.01 | 0.39 | **0.63** | 0.55 |
| IUE7 | Trying to commit suicide. | 8.33 | 19.17 | 0.25 | **0.51** | 0.33 |
| | Eigenvalues | | | **6.23** | 1.35 | |
| | % of explained variance | | | **44.49** | 9.66 | |
| | Cronbach's α | | | **0.88** | 0.82 | |

### 3.3.5. Belief in God and Level of Religiosity

Two simple multiple-choice questions were included in the survey: "Do you believe in God" (1 = I never believed; 2 = I don't believe it, but I already believed; 3 = Now I believe but I didn't believe before; 4 = I always believed) and "Do you consider yourself a religious person?" (Likert scale, from 1 = not religious to 5 = very religious).

### 3.4. Procedures

The questionnaires were administered by the author and a team of students as part of a research work of the curricular unit of Research Methods of a faculty from the University of Coimbra. The authors of this study provided training in survey data collection and ethical standards. Each student was invited to collect responses from one recurring player of games of chance and gambling (eligibility criteria). Participants were contacted by these students in person, by e-mail, or by telephone, and a date was agreed for the delivery of the questionnaire. Responses were anonymous and delivered in sealed envelopes, delivered by the research team. Anonymity and confidentiality of all participants and their personal answers were ensured for ethical reasons and to avoid biases in their answers.

The questionnaire began with an explanation of the study, clear instructions and guarantee of anonymity and confidentiality of answers, the voluntary nature of participation, and informed consent. The inclusion criterion was to be a recurring player of games of chance and gambling.

### 4. Results

According to Table 5, the majority of players believe in God ($M$ = 3.58) but do not consider themselves significantly religious ($M$ = 2.73). They showed moderate scores in the *Religious Beliefs and Attitudes Scale* ($M$ = 3.26) and are more optimistic ($M$ = 3.61) than pessimistic ($M$ = 2.58), $t(270)$ = 18.20, $p < 0.001$. On average, players estimate chances of 66% of intrinsic desirable events (e.g., having harmony in the family, living happily, overcoming the biggest difficulty) and fewer probabilities of occurring extrinsic desirable events ($M$ = 36%; e.g., become a millionaire, win the lottery, be famous), namely highly unlikely desirable events, $t(270)$ = 24.08, $p < 0.001$. Inversely, the average estimation of extrinsic undesirable events (e.g., dying soon, losing love, having a serious accident, going through difficult times) is higher than the estimation of intrinsic undesirable events (e.g., not achieving idealization, not being able to fulfill duties, losing hope, etc.), $t(270)$ = 15.98, $p < 0.001$.

The overall influence of religious beliefs and attitudes on optimism was positive although weak ($r$ = 0.14, shared variance of $R^2$ = 1.96%), as well as with pessimism ($r$ = 0.20, $R^2$ = 4.0%). The relationship between religious beliefs and attitudes was higher with the probability of extrinsic desirable events ($r$ = 0.27), indicating that the higher the level of religious beliefs, the more the person believes in the probability of occurrence of extrinsic desirable events (namely, become a millionaire, win the lottery, be famous, a miracle happens in life, and be admired by other people) with a proportion of shared variance of 7.29%. The association of this dimension of optimism it was also positive with the belief in God ($r$ = 0.17), and the level of religiosity ($r$ = 0.19%), although with lower magnitude (shared variances of 2.89% and 3.61%, respectively).

The probability of extrinsic desirable events showed positive correlations with the probability of both extrinsic and intrinsic undesirable events ($r$ = 0.32 and 0.23, $R^2$ = 10.24% and 5.29% of shared variance). Extrinsic optimism was more correlated with religious beliefs and attitudes in comparison with intrinsic optimism ($r$ = 0.27 vs. $r$ = 0.20, $R^2$ = 7.29% vs. $R^2$ = 4.0%). Religious beliefs and attitudes seems to be similarly correlated with extrinsic and intrinsic pessimism events ($r$ = 0.20 and 0.18, $R^2$ = 4.0% and 3.24% of shared variance).

**Table 5.** Descriptive statistics (min, max, Mean, SD) and intercorrelation matrix.

| | Min. | Max. | Mean | SD | 1 | 2 | 3 | 4 | 5 | 6 | 7 | 8 | 9 |
|---|---|---|---|---|---|---|---|---|---|---|---|---|---|
| 1. Religious Beliefs and Attitudes Scale (1 to 5 points) | 1.08 | 5.00 | 3.26 | 1.00 | 1 | 0.14 * | 0.23 ** | 0.20 ** | 0.27 ** | 0.20 ** | 0.18 ** | 0.71 ** | 0.69 ** |
| 2. Optimism (1 to 5 points) | 2.14 | 5.00 | 3.61 | 0.52 | | 1 | −0.15 * | 0.27 ** | 0.27 ** | 0.00 | −0.13 * | 0.08 | 0.10 |
| 3. Pessimism (1 to 5 points) | 1.00 | 4.40 | 2.58 | 0.71 | | | 1 | −0.09 | 0.07 | 0.26 ** | 0.34 ** | 0.16 ** | 0.04 |
| 4. F2: Intrinsic desirable events (probability 0–100%) | 0.50 | 100.00 | 65.98 | 19.28 | | | | 1 | 0.52 ** | 0.25 ** | 0.11 | 0.14 * | 0.16 ** |
| 5. F Extrinsic desirable events (probability 0–100%) | 0.07 | 100.00 | 36.02 | 22.32 | | | | | 1 | 0.32 ** | 0.23 ** | 0.17 ** | 0.19 ** |
| 6. Extrinsic undesirable events (probability 0–100%) | 0.00 | 100.00 | 42.87 | 21.69 | | | | | | 1 | 0.66 ** | 0.07 | 0.00 |
| 7. Intrinsic undesirable events (probability 0–100%) | 0.00 | 100.00 | 26.90 | 17.22 | | | | | | | 1 | 0.10 | 0.00 |
| 8. Do you believe in God (1 to 4 points) | 1.00 | 4.00 | 3.58 | 0.88 | | | | | | | | 1 | 0.58 ** |
| 9. Do you consider yourself a religious person? (1 to 5 points) | 1.00 | 5.00 | 2.73 | 0.90 | | | | | | | | | 1 |

* $p < 0.05$; ** $p < 0.01$.

Considering the specificity of the association between the global score of the *Religious Beliefs and Attitudes Scale* and the probability of winning the lottery or becoming a millionaire, we found a significant positive score just for winning the lottery ($r = 0.15$, $p = 0.015$), although the effect size is low ($R^2 = 2.25\%$ of shared variance). Furthermore, the correlation between the global score of the *Religious Beliefs and Attitudes Scale* and the probability of becoming a millionaire is not significant ($r = 0.09$, $p = 0.016$).

The influence of Education level was tested, considering four levels: 1 = until 4 years of school; 2 = until 9 years of school; 3 = until 12 years of school; and 4 = more than 12 years of school. An ANOVA (general linear model) was performed, taking Education as Independent Variable and the Religious Beliefs and Attitudes as the first Dependent Variable. We found an effect size of Education of 9.4%, $F (3, 267) = 9.20$, $p < 0.001$, $\eta_p^2 = 0.094$, $(1-\beta) = 0.996$. The post hoc tests Tukey HSD identified higher religious beliefs and attitudes in participants with fewer years of school. The effect of Education concerning the dimension *Optimism* was non-significant, $F(3, 267) = 1.87$, $p = 0.136$, $\eta_p^2 = 0.021$, $(1-\beta) = 0.481$. Considering *Pessimism* dimension, a significant difference was found, with an effect size of 6% [$F (3, 267) = 5.69$, $p = 0.001$, $\eta_p^2 = 0.060$, $(1-\beta) = 0.946$], due to the higher levels of pessimism in players with fewer years of education in comparison with players with higher education (mean difference of 0.45 and of 0.39 with 4 and 9 years of education, respectively, $p < 0.01$). For the *Estimation of Future Desirable Events Scale*, the MANOVA performed did not show any significant effect size for Education, Wilks' lambda = 0.960, $F (6, 532) = 1.84$, $p = 0.087$, $\eta_p^2 = 0.021$, $(1-\beta) = 0.690$. At last, for the *Estimation of Undesirable Events Scale*, the MANOVA showed a slight effect size for Education (2.4%), Wilks' lambda = 0.953, $F (6, 532) = 1.84$, $p = 0.047$, $\eta_p^2 = 0.024$, although with low observed power, $(1-\beta) = 0.766$. Attending to these results, we did not consider education level as covariate in the model.

The structural model of the influence of religious beliefs and attitudes (*Religious Beliefs and Attitudes Scale*, *belief in God*, and *level of religiosity*) on individuals' optimism and pessimism is shown in Figure 1. For the *Optimism* construct, we considered the items of the Optimism scale, as well as the items evaluating the probability of intrinsic and extrinsic desirable events. For the *Pessimism* construct, we considered the items of the Pessimism scale and the items evaluating the probability of intrinsic and extrinsic undesirable events. The fit index CMIN/DF = 1.94 obtained indicated a good model fit. With respect to the RMSEA, we found the 0.059 value (90CI of 0.055 to 0.062), considered as an acceptable fit indicator, as well as the NFI = 0.71 and the CFI = 0.84 scores. This model indicates that the religious beliefs and attitudes had a higher influence on the players' optimism ($R^2 = 44\%$ of explained variance, $\beta = 0.66$, $p < 0.001$) in comparison with the players' pessimism ($R^2 = 5\%$, $\beta = 0.23$, $p = 0.034$). Briefly, religiosity (religious beliefs and attitudes in our model), showed an effect of 44% in the prediction of optimism and only an effect of 5% in the prediction of pessimism.

The differentiation between intrinsic and extrinsic optimism and pessimism allows us to go further in the influence of religious beliefs and attitudes. The *Optimism and Pessimism Scale* was considered a measure of intrinsic optimism (*F1-Optimism*) and intrinsic pessimism (*F2-Pessimism*). Two additional structural models were built, one for optimism and another for pessimism.

Considering the structural model for *Optimism* (see Figure 2), two dimensions were considered: *Intrinsic Optimism* (operationalized with a latent variable composed of the *Optimism* factor of the *Optimism and Pessimism Scale* and the dimension *F1-Intrinsic desirable events* of the *Estimation of Future Desirable Events Scale*) and *Extrinsic Optimism* (latent variable composed of the *dimension F2-Extrinsic desirable events* of the *Estimation of Future Desirable Events Scale*). *Religious beliefs and attitudes* explain 12% of Intrinsic *Optimism* ($R^2$ 0.12; $\beta = 0.34$, $p = 0.056$), 5% of *Extrinsic undesirable events* ($R^2 = 0.05$, $\beta = 0.22$, $p = 0.008$), and 3.5% of *Intrinsic undesirable events* ($R^2 = 0.035$, $\beta = 0.065$ direct effect + $\beta = 0.177$ indirect effect). We obtained good fit indices for this model considering CMIN/DF=1.90, NFI = 0.831, and CFI = 0.912, and an acceptable fit attending to RMSEA = 0.058.

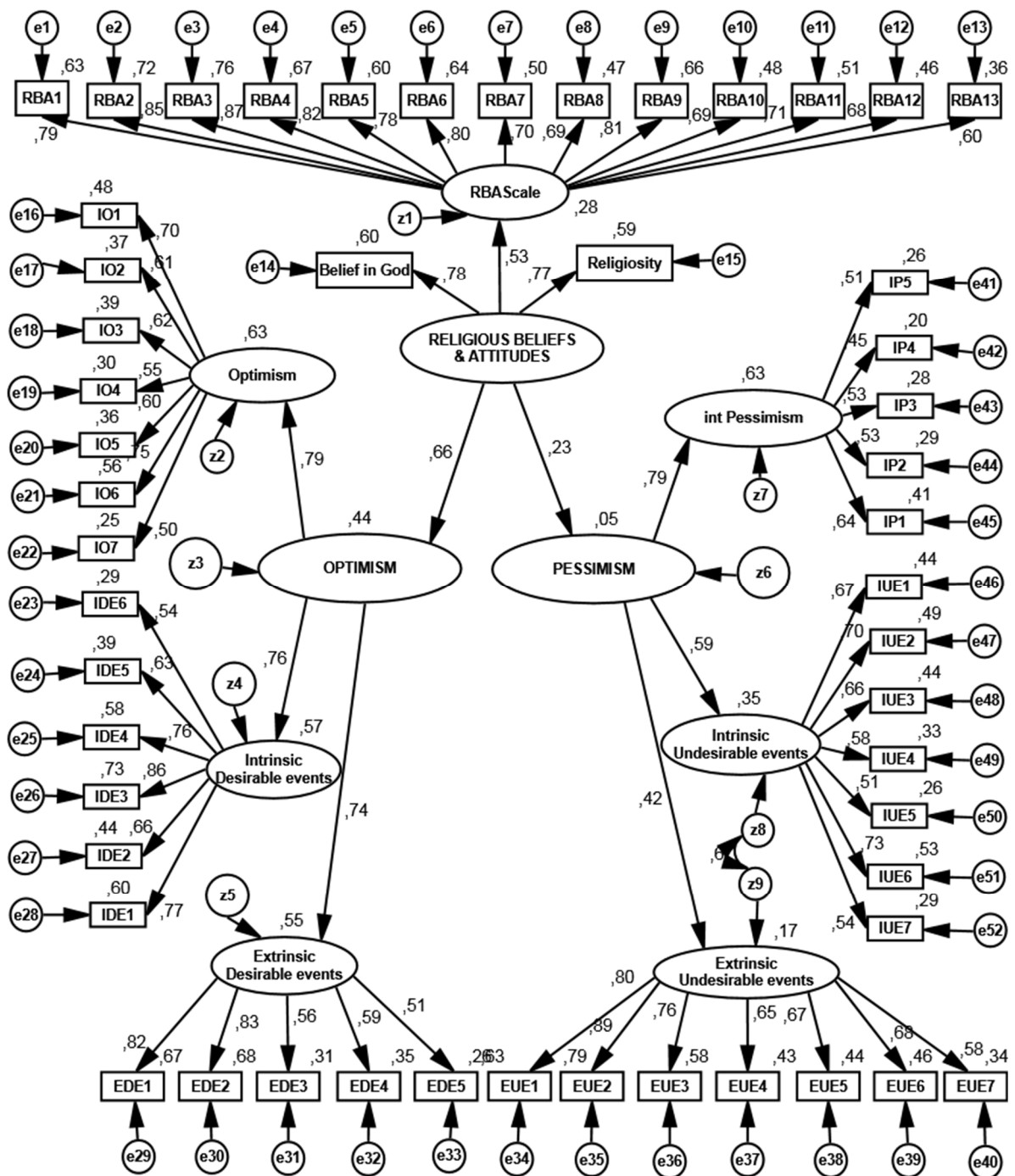

**Figure 1.** Influence of religious beliefs and attitudes on players' optimism and pessimism: Standardized regression coefficients and proportions of explained variance of the estimated structural model.

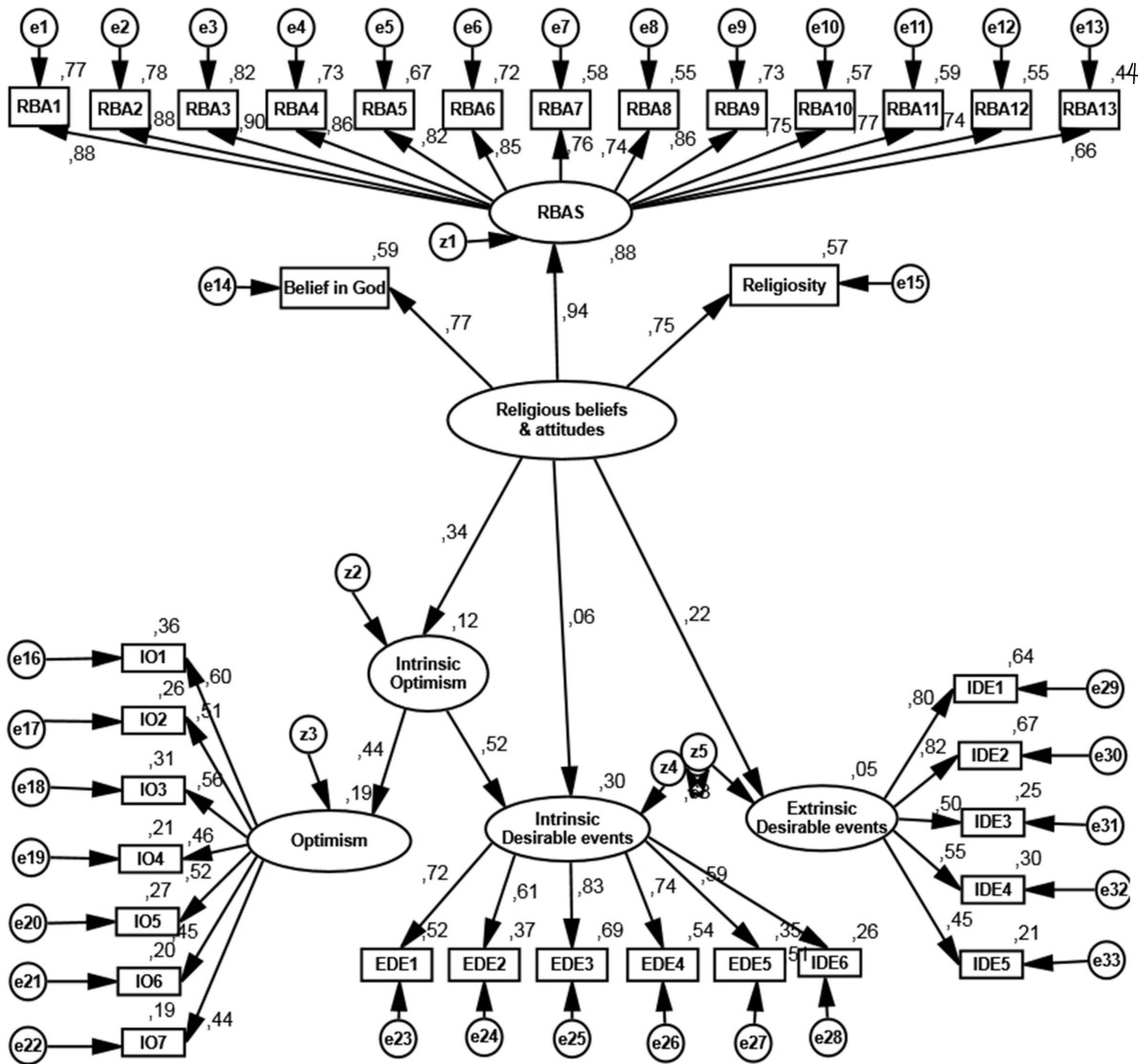

**Figure 2.** Influence of religious beliefs and attitudes on players' intrinsic and extrinsic optimism: Standardized regression coefficients and proportions of explained variance of the estimated structural model.

Attending to the structural model for *Pessimism* (see Figure 3), two dimensions were also considered: *Intrinsic Pessimism* (operationalized with a latent variable composed of the *Pessimism* factor of the *Optimism and Pessimism Scale* and the dimension *F1-Intrinsic undesirable events* of the *Estimation of Future Undesirable Events Scale*) and *Extrinsic Pessimism* (latent variable composed of the dimension F2-Extrinsic undesirable events of the *Estimation of Future Undesirable Events Scale*). *Religious beliefs and attitudes* explain 13% of *Intrinsic Pessimism* ($R^2 = 0.13$, $\beta = 0.37$, $p = 0.053$), 3.24% of *Extrinsic undesirable events* ($R^2 = 0.0324$, $\beta = 0.18$, $p = 0.008$), and 1.88% of Intrinsic undesirable events ($R^2 = 0.0188$, $\beta = 0.065$ direct effect + $\beta = 0.137$ indirect effect). We obtained good fit indices for this model considering CMIN/DF = 1.85, NFI = 0.840, and CFI = 0.919, and an acceptable fit attending to RMSEA = 0.056.

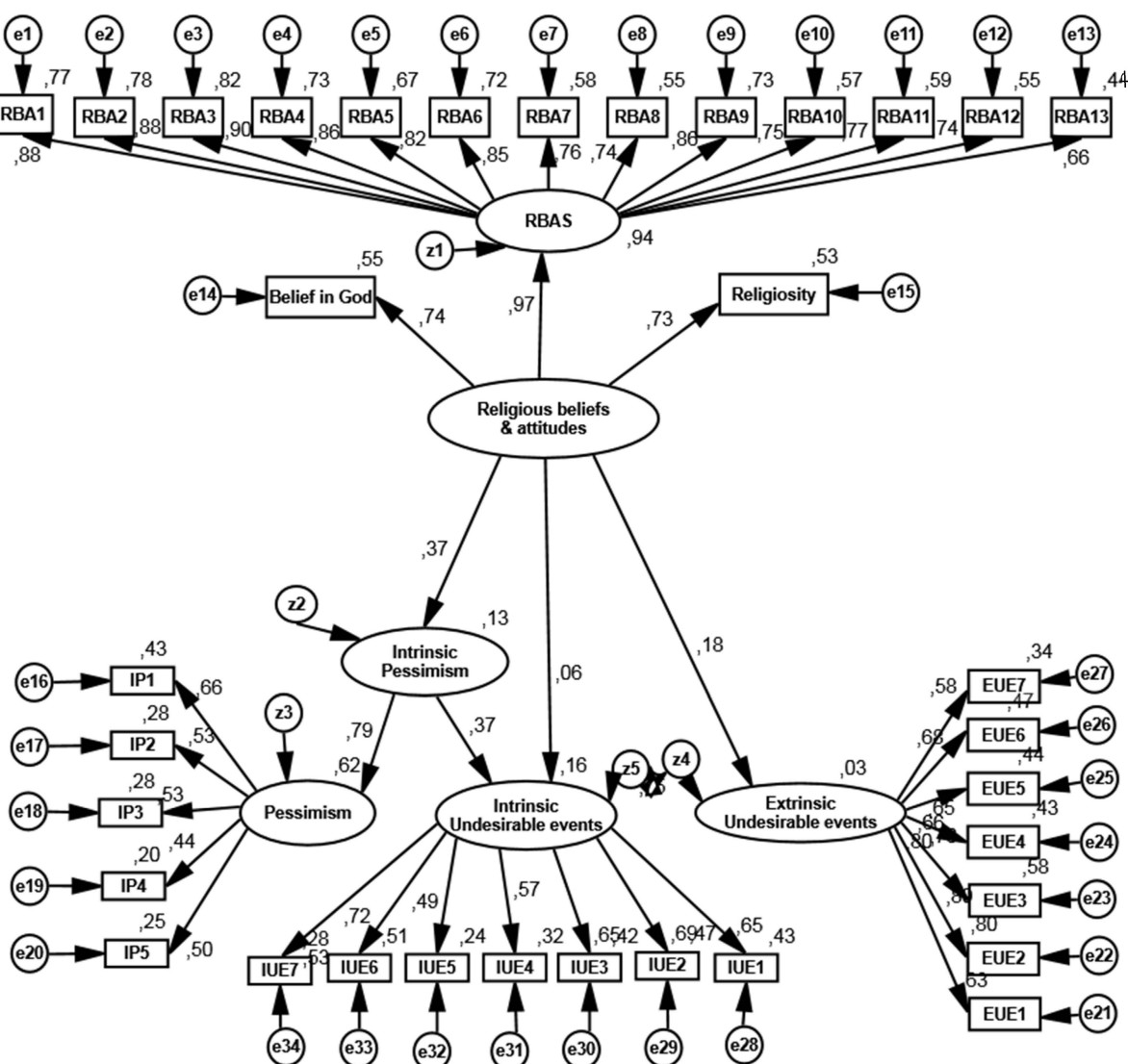

**Figure 3.** Influence of religious beliefs and attitudes on players' intrinsic and extrinsic pessimism: Standardized regression coefficients and proportions of explained variance of the estimated structural model.

## 5. Dicussion and Conclusions

Human beings are characterized by a permanent awareness that they are individualized beings, with their own existence distinct from others. Considering the circumstances of each situation, individuals tend to be more or less optimistic. Given the nature of optimism and the fact that it has always been considered a positive, strong, and general distortion in self-benefit (Armor and Taylor 1998; Carver and Scheier 2001; Domino and Conway 2001; Scheier and Carver 1985, 1992), with this research we aimed to understand the agentic features of this construct in the religiousness of game players of chance and gambling.

"Optimism is seen as a cognitive feature (a goal, an expectation, a belief or a causal attribution) about the desired and perceived as successful future" (Barros 2004, p. 101), entailing an involvement with the uncertainty factor. The data collected from the players of our sample enable us to ascertain to what extent dealing with risk is associated with optimism, as well as to what extent religiosity acts as a catalyst for this optimism in situations of games of chance and gambling.

Many studies have shown that, in general, people have expectations of positive results for their own lives. Widely known for optimism, this phenomenon was found in a variety

of situations, in samples of all ages and different cultures, Western and non-Western. It is consistent over time and, despite some specificities, during the events.

In this research, the constructs of optimism and pessimism emerged as distinct, tending to be independent rather than inversely related. However, players were more optimistic than pessimistic and more likely to estimate intrinsically desirable events (e.g., living happily, overcoming the biggest difficulty) in comparison with extrinsically desirable events (e.g., becoming a millionaire, being famous).

Considering the nature of optimism and the fact that it has always been characterized as a generalized and robust phenomenon (Domino and Conway 2001), in general terms, we can say that, in addition to people perceiving their future as being more positive than the from others (believing that they are more likely to experience desirable situations and less likely to experience undesirable events) in a variety of circumstances, and compared to those others, they believe they are superior. Given the very low probabilities of winning the lottery or other games of chance, this self-serving bias may explain the high probabilities that players of our sample indicated of winning the lottery (45% in our sample). In circumstances where any kind of evidence—objective in nature or via social comparison—indicates poor probabilities of personal success, such as winning the lottery, individual beliefs play a key role in maintaining the levels of idiosyncratic optimism (Mónico 2011, 2021; Perloff and Fetzer 1986). Among these beliefs, this research was dedicated to those of a religious nature, due to the lack of studies on the religiosity–optimism interconnection, especially in players of games of chance and gambling. Our results show a positive influence of religiosity on optimism.

Given the unpredictability of the situations, individuals are led to develop adaptive strategies in order to maintain or recover the perception of control, including the creation of illusory beliefs of control (Taylor and Brown 1999). Such beliefs are positively associated with subjective well-being and the ability to adjust to threatening and unpredictable situations (for instance, Diez-Esteban et al. 2019, found an influence of the religious backgrounds on corporate risk-taking); these situations are controlled by cognitive strategies, characterized by patterns of religious, political, and/or technological control beliefs. In fact, several studies with people living in an uncontrollable threat situation show that those who show signs of greater psychological well-being and better adjustment to the threat situation are the ones who have developed illusions of control over this threat. Those who have these secondary control schemes automatically activate them in threatening situations, reducing the insecurity and anxiety of the situation and, in this way, restoring the feeling of well-being.

A considerable part of individuals' religiosity satisfies control needs and, in the specific case of the games of chance and gambling, can act as an illusion of control. McCullough and Willoughby (2009) present six key propositions that interrelate religion, self-regulation, and self-control: "(a) that religion can promote self-control; (b) that religion influences how goals are selected, pursued, and organized; (c) that religion facilitates self-monitoring; (d) that religion fosters the development of self-regulatory strength; (e) that religion prescribes and fosters proficiency in a suite of self-regulatory behaviors; and (f) that some of religion's influences on health, well-being, and social behavior may result from religion's influences on self-control and self-regulation" (p. 69).

Divided into the ideological, intellectual, consequential, and ritualistic dimensions (Glock and Stark 1965), religiousness as a propeller of the optimism in players has become the outline of this research. In general, we can conclude that religious beliefs and attitudes are positively associated with optimism in these kinds of players. However, and surprisingly, results show as well a positive association with pessimism, although with a lower effect size. It seems that both optimism and pessimism are evident in these types of players and that they estimate an increased probability of occurrence of positive events but also of negative events. In fact, despite the optimism levels in players of games of chance of our sample, they also presented higher probabilities for undesirable events, especially for extrinsic undesirable events (approx. 43% in our sample), like having a serious chronic or

malignant disease or having a serious accident. These results seem to indicate that players of games of chance estimate increased probabilities of both positive and negative events occurring in their lives, supporting the idea that optimism and pessimism are independent constructs. Indeed, research concerning pessimism and, in particular, comparative pessimism (Kruger and Burrus 2004; Taylor and Shepperd 1998) refute the omnipresence of the optimistic trait. Desirable unusual and common undesirable occurrences encourage pessimism, which leads us to deduce that optimism is not a phenomenon as robust and widespread as the literature seems to show (Mónico 2011, 2021). Our results are in line with this finding, or rather, that individuals who are very optimistic towards some situations can also be very pessimistic concerning other situations.

Searching for an interpretation for these results, in recent decades researchers in positive psychology have come to recognize self-regulation as an important aspect of the self, such as resilience, adaptation to adversities (Barros 2004; Brown 1987; Higgins et al. 1999), or even spiritual and religious development (McCullough and Boker 2007; Pargament and Mahoney 2021). We consider that individuals can use beliefs and religious behaviors as a self-regulatory mechanism, which confers them some stability and promotes optimism. As McCullough and Boker (2007) state, "To some extent, spiritual and religious changes may also be caused by self-regulation processes that are intrinsic to individual functioning" (p. 385). The importance that each one gives religion is, in some way, regulated by the functioning of an internal orientation system that seeks to achieve internal balance.

Additionally, the distinction between intrinsic and extrinsic optimism in our sample has shown that the players anchor their optimism in different dimensions of beliefs. The belief that faith or a given way of connecting to the sacred or the divinity(s)—in short, spirituality (Barros 2000; Mónico 2021)—helps the individual to achieve the desired optimism, believing that they can win. The belief in the spiritual, integrated into the intrinsic dimension of the Allportian sense of religious experience (Allport 1966) depends, however, on an external factor: the action of the divine in the subject. In our research, we complement with the attribute of extrinsic the optimism that is based on factors external to oneself (i.e., the belief that the desired results are dependent on a supernatural will or intervention); on the other hand, we base intrinsic optimism when it refers to the positive disposition or attitude that the good results are directly dependent on the individual's aptitudes (Carver and Scheier 2014).

People tend to perceive the world as controllable, revealing the perception of control over the surrounding environment as a need for each individual (Wegener and Bargh 1998). If it is a truism to mention that human beings are faced daily with the unexpected in their environment, perhaps it is not to say that control beliefs, among which we highlight those of a religious scope, are inscribed in the skills developed by each individual with a view to adaptation to the environment (Taylor and Brown 1999). According to McCullough and Willoughby (2009), religiosity constitutes a means of activating self-control and, as evidenced by Buchanan and Seligman (1995), each person may be situated on a continuum that distances from extreme internality to externalities. By integrating this dichotomization in the individual cognitive style regarding the modes of information processing in relation to positive future expectations, we can elaborate for optimism a line of reasoning analogous to the one established for the locus of control (Rotter 1990)—internal vs. external. We consider that an individual whose optimism (or pessimism) is based on beliefs of internality expects the good (or bad) future experiences to depend on his or her own personal aptitudes (or lack of them); the positive (or negative) expectations they maintain are formulated based on the expected results of personal actions. On the other hand, those whose optimism (or pessimism) is based on external factors tends to maintain the conviction that good (or bad) results will prevail due to situational factors, not exercising control over these factors; in this situations, individual optimism or pessimism is not centered on personal factors but, on the contrary, on factors outside the self such as luck, chance or even the help of some (super)natural entity.



Our results evidence that extrinsic desirable events, like winning the lottery, were more predicted by religious beliefs and attitudes in comparison with intrinsic desirable events. In reverse, religious beliefs and attitudes tend to slightly predict more intrinsic pessimism in comparison with intrinsic optimism. Concluding, our results demonstrate the importance of distinguishing internal causes from external causes in the kind of beliefs underlying optimism and pessimism. We believe that we can find a continuous distribution of optimists and pessimists between the two extremes, that is, between those who base their optimism (or their pessimism) solely on factors of an internal nature and those that cement it entirely on externality.

At last, the present research has some limitations that should be addressed. The variables used in this study allowed us to analyze the effect of religious beliefs and attitudes in optimism and pessimism dimensions in players of games of chance and gambling. Although these outputs relating to religiosity, optimism, and pessimism are positive and promising, it is important to replicate this research by introducing new measures, especially focused on intrinsic and extrinsic optimism and pessimism. In relation to the sample, additional studies should be considered, comparing gamblers with no gamblers. The research can also be extended to other players, differentiating results according to the frequency of playing and introducing control variables like the addition games.

**Author Contributions:** Conceptualization, L.S.M. and V.R.A.; methodology, L.S.M.; software, V.R.A. and L.S.M.; validation, L.S.M.; formal analysis, L.S.M.; investigation, L.S.M.; resources, L.S.M. and V.R.A.; data curation, L.S.M. and V.R.A.; writing—original draft preparation, L.S.M.; writing—review and editing, L.S.M. and V.R.A. All authors have read and agreed to the published version of the manuscript.

**Funding:** This research received no external funding.

**Informed Consent Statement:** Informed consent was obtained from all subjects involved in the study.

**Data Availability Statement:** The data presented in this study are available on request from the first author, e-mail: lisete.monico@fpce.uc.pt.

**Acknowledgments:** We acknowledge all the participants in this study, and also the person involved in data collection.

**Conflicts of Interest:** The authors declare no conflict of interest.

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
