# Peer review of "The Effect of Religious Beliefs and Attitudes in Intrinsic and Extrinsic Optimism and Pessimism in Players of Games of Chance"

_religions, doi:10.3390/rel13020097_

Round 1
Reviewer 1 Report
Is it possible to make the results more intelligible by predicting the effect of a change? For example, a person who is 10% more religious is X% more optimistic. Just a few sentences like this would make the results more intelligible to a wider range of readers.
It does not appear that the results are taking account of the age and education of the respondents. It seems to me that better educated respondents are less likely to overestimate the probability of good events, and less likely to associate them with religion. In other words, the results may vary by education (and/or IQ).
In the conclusion, the authors talk about expanding the study to include other players, players of different playtime intensity, different games, etc. I think the larger point is, we have no baseline in this study against which to assess the connections found here. What is the relationship between optimism and religiosity among people who are not gamblers?
Author Response
Point 1: Is it possible to make the results more intelligible by predicting the effect of a change? For example, a person who is 10% more religious is X% more optimistic. Just a few sentences like this would make the results more intelligible to a wider range of readers.
Response 1: Thank you for the suggestion. We improved the Results section in order to to make the results more intelligible by predicting the effect of a change. Namely,
- it was added on the second paragraph of the Results section the following explanation: “The relationship between religious beliefs and attitudes was higher with the probability of extrinsic desirable events (r = .27), indicating that the higher the level of religious beliefs, the more the person believes in the probability of occurrence of extrinsic desirable events - namely, become a millionaire, win the lottery, be famous, a miracle happens in life, and be admired by other people -, with a proportion of shared variance of 7.29%. The association of this dimension of optimism it was also positive with the belief in God (r = .17), and the level of religiosity (r = .19 %), although with lower magnitude (shared variances of 2.89% and 3.61%, respectively).”
- it was added on the first paragraph of the Results section after Table 5 the following explanation: “Briefly, religiosity (religious beliefs and attitudes in our model), showed an effect of 44% in the prediction of optimism and only an effect of 5% in the prediction of pessimism.”
- it was added on the first paragraph of the Results section after Figure 1 the following explanation: “Religious beliefs and attitudes explain 12% of Intrinsic Optimism (..), 5% of Extrinsic undesirable events (…), and), 3.5% of Intrinsic undesirable events (…).”
- it was added on the first paragraph of the Results section after Figure 2 the following explanation: “Religious beliefs and attitudes explain 13% of Intrinsic Pessimism (…), 3.24% of Extrinsic undesirable events (…), and 1.88% of Intrinsic undesirable events (…).
Point 2: It does not appear that the results are taking account of the age and education of the respondents. It seems to me that better educated respondents are less likely to overestimate the probability of good events, and less likely to associate them with religion. In other words, the results may vary by education (and/or IQ).
Response 2: We checked for the differences between education (we did not collect data from IQ), considering four levels (1 = until 4 years of school; 2 = until 9 years of school; 3 = until 12 years of school; and 4 = more than 12 years of school), and run an Analysis of Variance (ANOVA, general linear model), taking Education as Independent Variable and the Religious Beliefs and Attitudes as Dependent Variable. We found a effect size of Education of 9.4%, F (3, 267) = 9.20, p < .001, ηp2 = .094, (1-β) = .996. The post-hoc tests Tukey HSD identified higher religious beliefs and attitudes in participants with fewer years of school. However, the effect size of the differences of Education level concerning the dimension Optimism was non significant, F (3, 267) = 1.87 , p = .136, ηp2 = .021, (1-β) = .481 – with none significant pairwise differences identified through the Tukey HSD tests. Considering Pessimism a significant difference was found, with an effect size of 6% [F (3, 267) = 5.69, p = .001, ηp2 = .060, (1-β) = .946], due to the higher levels of pessimism found in the participants with fewer years of education in comparison with players with higher education (mean difference of 0.45 and of 0.39 with 4 and 9 years of education, respectively, p < .01).
For the Estimation of Future Desirable Events Scale, the MANOVA performed didn’t show any significant effect size for Education, Wilks’ lambda = .960, F (6, 532) = 1.84, p = .087, ηp2 = .021, (1-β) = .690. At last, for the Estimation of Undesirable Events Scale, the MANOVA showed a slight effect size for Education (2.4%), Wilks’ lambda = .953, F (6, 532) = 1.84, p = .047, ηp2 = .024, although with low observed power, (1-β) = .766. For all these reasons we didn´t insert the education level in the model as covariate.
The following note was added to the manuscript, before Table 5:
“The influence of Education level was tested, considering four levels: 1 = until 4 years of school; 2 = until 9 years of school; 3 = until 12 years of school; and 4 = more than 12 years of school. An ANOVA (general linear model) was performed, taking Education as Independent Variable and the Religious Beliefs and Attitudes as the first Dependent Variable. We found an effect size of Education of 9.4%, F (3, 267) = 9.20, p < .001, ηp2 = .094, (1-β) = .996. The post-hoc tests Tukey HSD identified higher religious beliefs and attitudes in participants with fewer years of school. The effect of Education concerning the dimension Optimism was non significant, F(3, 267) = 1.87 , p = .136, ηp2 = .021, (1-β) = .481. Considering Pessimism dimension, a significant difference was found, with an effect size of 6% [F (3, 267) = 5.69, p = .001, ηp2 = .060, (1-β) = .946], due to the higher levels of pessimism in players with fewer years of education in comparison with players with higher education (mean difference of 0.45 and of 0.39 with 4 and 9 years of education, respectively, p < .01). For the Estimation of Future Desirable Events Scale, the MANOVA performed didn’t show any significant effect size for Education, Wilks’ lambda = .960, F (6, 532) = 1.84, p = .087, ηp2 = .021, (1-β) = .690. At last, for the Estimation of Undesirable Events Scale, the MANOVA showed a slight effect size for Education (2.4%), Wilks’ lambda = .953, F (6, 532) = 1.84, p = .047, ηp2 = .024, although with low observed power, (1-β) = .766. Attending to these results, we didn´t consider education level as covariate in the model.”
We also added this paragraph in the data analysis section (3.2)
“For the analysis of variance (ANOVA) and multivariate analysis of variance (MANOVA), the assumptions of independence of observations and homogeneity of error variance and covariance matrices of the dependent variables were checked. Post-hoc Tukey HSD tests were performed for pairwise multiple comparisons.”
Point 3: In the conclusion, the authors talk about expanding the study to include other players, players of different playtime intensity, different games, etc. I think the larger point is, we have no baseline in this study against which to assess the connections found here. What is the relationship between optimism and religiosity among people who are not gamblers?
Response 3: Thank you for this suggestion, very pertinent to include in further studies. Indeed, we have made research using this measures with other samples (elderly vs youth, believers vs atheists, healthy vs sick), but in these samples we did not ask for the player/game questions. We included the reviewer’s suggestion in the last paragraph of the paper:
“In relation to the sample, additional studies should be considered, comparing gamblers with no gamblers. The research can also be extended to other players, differentiating results according to the frequency of playing and introducing control variables like the addition games.”
Reviewer 2 Report
Thank you for the opportunity to review this interesting article.
The research is essential to the field and I consider it as novel.
Regarding the bibliographic support, the references used, in a general way, are very old For instance: specifically to the definition of religiosity, the most recent reference is from 2010.
The authors provide a detailed description of the sample. However, it is pertinent to ask: regarding the procedures, how many emails were sent and how many returned? How many of them were valid?
The methodology is correctly used and the description provided of all procedures is adequate. However, what are the theoretical and practical contributions of the work? Considering the theme of the issue, much more could be said about it, in order to make the article more robust and underline the novelty.
It were detected some inconsistencies in the English language, as well as, formatting problem of the document; both of these problems must be improved.
Congratulations for your research and article proposal.
Author Response
Point 1: Thank you for the opportunity to review this interesting article. The research is essential to the field and I consider it as novel. Regarding the bibliographic support, the references used, in a general way, are very old For instance: specifically to the definition of religiosity, the most recent reference is from 2010.
Response 1: Thank you for revising our paper and for your notes. We opted by using classic literature in the field of religiosity and optimism, balancing it with more recent publications in the field. However, we noticed that the Editorial Office removed all the references from the paper, in order to guarantee authors’ anonymity. Throughout the text, in some occasions it appears the following information: “omited author for peer review”. This references ommited are more updated (namely, we have one book of 2021, one paper of 2019, one paper of 2016, some papers of 2013 and 2012).
Additionally, we added in this revision the following references, more updated for the domain of religiosity/spirituality/religion:
Diez-Esteban JM, Farinha JB, and Garcia-Gomez CD. 2019. Are religion and culture relevant for corporate risk-taking? International evidence. Business Research Quality, 22, 1: 36–55. https://doi.org/10.1016/j.brq.2018.06.003.
Sitzmann, Traci; and Campbell, Elizabeth M. The hidden cost of prayer: religiosity and the gender wage GAP. Academy of Management Journal, 64,4: 1016-1048. 10.5465/amj.2019.1254.
Martins, Ricardo C. 2021. Religiosidade no século XXI: misticismo, ateísmo ou indiferença. Revista sem Aspas, 10, e02100. https://doi.org/10.29373/sas.v10i00.15330
Mónico, Lisete S., and Margaça, Clara 2021. The Workaholism Phenomenon in Portugal: Dimensions and Relations with Workplace Spirituality. Religions 12: 852. https://doi.org/10.3390/rel12100852
We also added the following reference, converning data analysis:
Tabachnick, Barbara G., and Fidell, Linda S. 2019. Using multivariate statistics (7th ed.) New Jersey: Pearson Education.
Point 2: The authors provide a detailed description of the sample. However, it is pertinent to ask: regarding the procedures, how many emails were sent and how many returned? How many of them were valid?
Response 2: our data was collected in person with the cooperation of students trained in the practical classes of the subject of Research Methods of the Psychology degree. The authors of this study provided training in survey data collection and ethical standards. Each student was invited to collect responses from one recurring player of games of chance and gambling. Responses were anonymous and delivered in sealed envelopes.
We added the following paragraph to the article, in the point 3.4 Procedures:
The authors of this study provided training in survey data collection and ethical standards. Each student was invited to collect responses from one recurring player of games of chance and gambling (eligibility criteria). Participants were contacted by these students in person, by e-mail, or by telephone, and a date was agreed for delivery of the questionnaire. Responses were anonymous and delivered in sealed envelopes, delivered by the research team.
Point 3: The methodology is correctly used and the description provided of all procedures is adequate. However, what are the theoretical and practical contributions of the work? Considering the theme of the issue, much more could be said about it, in order to make the article more robust and underline the novelty.
Response 3: We improved the paper in order to to make the results more intelligible. Namely,
- it was added on the second paragraph of the Results section the following explanation: “The relationship between religious beliefs and attitudes was higher with the probability of extrinsic desirable events (r = .27), indicating that the higher the level of religious beliefs, the more the person believes in the probability of occurrence of extrinsic desirable events - namely, become a millionaire, win the lottery, be famous, a miracle happens in life, and be admired by other people -, with a proportion of shared variance of 7.29%. The association of this dimension of optimism it was also positive with the belief in God (r = .17), and the level of religiosity (r = .19 %), although with lower magnitude (shared variances of 2.89% and 3.61%, respectively).”
- it was added on the first paragraph of the Results section after Table 5 the following explanation: “Briefly, religiosity (religious beliefs and attitudes in our model), showed an effect of 44% in the prediction of optimism and only an effect of 5% in the prediction of pessimism.”
- it was added on the first paragraph of the Results section after Figure 1 the following explanation: “Religious beliefs and attitudes explain 12% of Intrinsic Optimism (..), 5% of Extrinsic undesirable events (…), and), 3.5% of Intrinsic undesirable events (…).”
- it was added on the first paragraph of the Results section after Figure 2 the following explanation: “Religious beliefs and attitudes explain 13% of Intrinsic Pessimism (…), 3.24% of Extrinsic undesirable events (…), and 1.88% of Intrinsic undesirable events (…).
- We checked for the differences between education (we did not collect data from IQ), considering four levels (1 = until 4 years of school; 2 = until 9 years of school; 3 = until 12 years of school; and 4 = more than 12 years of school), and run an Analysis of Variance (ANOVA, general linear model), taking Education as Independent Variable and the Religious Beliefs and Attitudes as Dependent Variable. We found a effect size of Education of 9.4%, F (3, 267) = 9.20, p < .001, ηp2 = .094, (1-β) = .996. The post-hoc tests Tukey HSD identified higher religious beliefs and attitudes in participants with fewer years of school. However, the effect size of the differences of Education level concerning the dimension Optimism was non significant, F (3, 267) = 1.87 , p = .136, ηp2 = .021, (1-β) = .481 – with none significant pairwise differences identified through the Tukey HSD tests. Considering Pessimism a significant difference was found, with an effect size of 6% [F (3, 267) = 5.69, p = .001, ηp2 = .060, (1-β) = .946], due to the higher levels of pessimism found in the participants with fewer years of education in comparison with players with higher education (mean difference of 0.45 and of 0.39 with 4 and 9 years of education, respectively, p < .01).
- For the Estimation of Future Desirable Events Scale, the MANOVA performed didn’t show any significant effect size for Education, Wilks’ lambda = .960, F (6, 532) = 1.84, p = .087, ηp2 = .021, (1-β) = .690. At last, for the Estimation of Undesirable Events Scale, the MANOVA showed a slight effect size for Education (2.4%), Wilks’ lambda = .953, F (6, 532) = 1.84, p = .047, ηp2 = .024, although with low observed power, (1-β) = .766. For all these reasons we didn´t insert the education level in the model as covariate.
- The following note was added to the manuscript, before Table 5:
“The influence of Education level was tested, considering four levels: 1 = until 4 years of school; 2 = until 9 years of school; 3 = until 12 years of school; and 4 = more than 12 years of school. An ANOVA (general linear model) was performed, taking Education as Independent Variable and the Religious Beliefs and Attitudes as the first Dependent Variable. We found an effect size of Education of 9.4%, F (3, 267) = 9.20, p < .001, ηp2 = .094, (1-β) = .996. The post-hoc tests Tukey HSD identified higher religious beliefs and attitudes in participants with fewer years of school. The effect of Education concerning the dimension Optimism was non significant, F(3, 267) = 1.87 , p = .136, ηp2 = .021, (1-β) = .481. Considering Pessimism dimension, a significant difference was found, with an effect size of 6% [F (3, 267) = 5.69, p = .001, ηp2 = .060, (1-β) = .946], due to the higher levels of pessimism in players with fewer years of education in comparison with players with higher education (mean difference of 0.45 and of 0.39 with 4 and 9 years of education, respectively, p < .01). For the Estimation of Future Desirable Events Scale, the MANOVA performed didn’t show any significant effect size for Education, Wilks’ lambda = .960, F (6, 532) = 1.84, p = .087, ηp2 = .021, (1-β) = .690. At last, for the Estimation of Undesirable Events Scale, the MANOVA showed a slight effect size for Education (2.4%), Wilks’ lambda = .953, F (6, 532) = 1.84, p = .047, ηp2 = .024, although with low observed power, (1-β) = .766. Attending to these results, we didn´t consider education level as covariate in the model.”
- We also added this paragraph in the data analysis section (3.2)
“For the analysis of variance (ANOVA) and multivariate analysis of variance (MANOVA), the assumptions of independence of observations and homogeneity of error variance and covariance matrices of the dependent variables were checked. Post-hoc Tukey HSD tests were performed for pairwise multiple comparisons.”
- In the Conclusions section, it was included in the last paragraph: “In relation to the sample, additional studies should be considered, comparing gamblers with no gamblers. The research can also be extended to other players, differentiating results according to the frequency of playing and introducing control variables like the addition games.”
Point 4: It were detected some inconsistencies in the English language, as well as, formatting problem of the document; both of these problems must be improved. Congratulations for your research and article proposal.
Response 4: The paper was revised considering spelling and grammar. Thank you for reading our paper and for the opportunity to improve it.
